# A Novel CdS Quantum Dots Decorated 3D Bi$_2$O$_2$CO$_3$ Hierarchical Nanoflower with Enhanced Photocatalytic Performance

**Zhilin Ji, Hongwei Wang * and Xilin She**

School of Environmental Science and Engineering, Qingdao University, Qingdao 266071, China;
jzl18205452955@163.com (Z.J.); xlshe@qdu.edu.cn (X.S.)
*   Correspondence: hwwang6688@qdu.edu.cn

**Abstract:** Heterojunction engineering has shown great potential in the field of photocatalysis to deal with environmental pollutants. The design and synthesis of heterojunction photocatalysts with high efficiency and stability still face great challenges. In this work, a novel CdS quantum dots (QDs) decorated 3D Bi$_2$O$_2$CO$_3$ hierarchical nanoflower heterojunction photocatalyst (Bi$_2$O$_2$CO$_3$/CdS QDs) was synthesized to investigate the photocatalytic Rhodamine B (RhB) degradation performance. CdS QDs were evenly distributed on the surface of the Bi$_2$O$_2$CO$_3$ nanoflower. Bi$_2$O$_2$CO$_3$/CdS QDs showed significantly enhanced photocatalytic RhB degradation performance compared with pristine Bi$_2$O$_2$CO$_3$ and CdS QDs. The enhanced photocatalytic performance was attributed to the synergistic effect of hierarchical structure and heterojunction, which greatly increased the active sites of the reaction and the photogenerated carriers transfer.

**Keywords:** photocatalysis; Bi$_2$O$_2$CO$_3$/CdS QDs; hierarchical structure; heterojunction

## 1. Introduction

As a green technology, photocatalysis has great potential in dealing with environmental pollution and the energy crisis. Photocatalyst is the key and core in the process of photocatalytic reaction, which can effectively convert solar energy into electrical energy and chemical energy [1–4]. However, the absorption and utilization of sunlight, the migration rate of photocarriers, the redox ability of active radicals and the number of reactive sites directly affect the photocatalytic performance of photocatalysts [5–8]. Therefore, it is of great significance to improve the photocatalytic performance by selecting appropriate photocatalyst and further modifying it.

Heterojunction engineering can effectively improve the photocatalytic performance of composite photocatalysts, and accelerate the transport of photocarriers in the photocatalytic process [9–11]. The three-dimensional (3D) hierarchical structures can bring advantages for the establishment of heterojunctions, and provide high specific surface area for the smooth progress of photocatalytic reaction. Huang et al. [12] successfully prepared a novel 3D WO$_{3-x}$/Bi/BiOCl hierarchical heterostructure to investigate the drastically enhanced photocatalytic degradation performance for RhB. They attributed the enhanced photocatalytic activity to the formed heterojunction structure and improved light-harvesting ability by 3D WO$_{3-x}$. Zhang et al. [13] fabricated a 3D Bi$_2$MoO$_6$/ZnO hierarchical heterostructured photocatalyst to explore the enhanced photocatalytic reduction performance for Cr(VI) under visible-light irradiation. The formation of the heterojunction between Bi$_2$MoO$_6$ and ZnO effectively facilitated the separation and transfer of electrons and holes, which enhanced the photocatalytic activity. Wang et al. [14] synthetized Ag and CdS quantum dots (QDs) co-decorated 3D hierarchical Bi$_5$O$_7$I heterojunction photocatalyst. The formation of tandem heterojunctions favored the

separation of charge carriers; the 3D hierarchical structure of $Bi_5O_7I$ offered adequate surface-active sites; and the surface plasmon resonance effect of Ag promoted the photothermal effect. These all endowed the $Bi_5O_7I/Ag/CdS$ photocatalyst with super photocatalytic activity. Based on the above, the fabrication of a 3D hierarchical photocatalyst with efficient heterojunction is beneficial to improve the photocatalytic activity.

Bismuth subcarbonate ($Bi_2O_2CO_3$) is an attractive photocatalyst for degrading environmental pollutants due to its low toxicity and good photocatalytic performance [15,16]. However, the wide band gap and the fast recombination of electrons and holes will limit its utilization value [17,18]. Morphology control and heterojunction can promote the absorption and utilization of solar energy, and accelerate the transmission of photogenerated carriers [5,19,20], thus improving the photocatalytic performance of $Bi_2O_2CO_3$. Herein, we fabricated a novel CdS QDs decorated 3D $Bi_2O_2CO_3$ hierarchical nanoflower photocatalyst ($Bi_2O_2CO_3$/CdS QDs) to investigate the photocatalytic Rhodamine B (RhB) degradation performance. CdS QDs are evenly distributed on the surface of the $Bi_2O_2CO_3$ nanoflower to form the heterojunction with $Bi_2O_2CO_3$ nanosheet. $Bi_2O_2CO_3$/CdS QDs displays significantly enhanced photocatalytic performance for RhB degradation. The 3D hierarchical structure provides a large number of active sites and the establishment of heterojunction accelerates the transmission of photogenerated carriers. The synergistic effect of hierarchical structure and heterojunction significantly improved the photocatalytic RhB degradation performance of $Bi_2O_2CO_3$/CdS QDs.

## 2. Results and Discussion

Figure 1 displays SEM images of the prepared photocatalysts. As shown in Figure 1A, the prepared $Bi_2O_2CO_3$ photocatalyst presents a nanoflower-like sphere with hierarchical structure, which can facilitate the deposition of CdS QDs. The pristine CdS agglomerates together, and presents the nanoparticle with the size of 60–100 nm shown in Figure 1B. As can be found from Figure 1C–F, the $Bi_2O_2CO_3$ hierarchical nanoflowers keep good flower shape after deposited with different amounts of CdS QDs. Some fragments are observed in the SEM images of $Bi_2O_2CO_3$/CdS QDs, which may be ascribed to the damage of $Bi_2O_2CO_3$ nanoflowers caused by ultrasound and agitation during the synthesis process of the heterojunction photocatalysts.

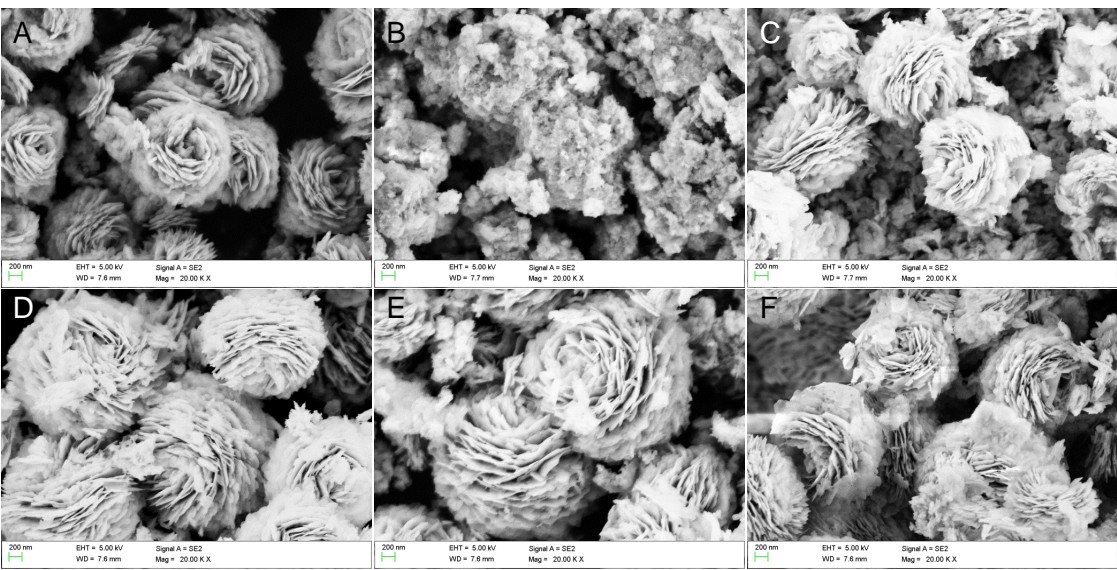

**Figure 1.** SEM images of the prepared photocatalysts: $Bi_2O_2CO_3$ (**A**), CdS QDs (**B**), $Bi_2O_2CO_3$/CdS QDs-5 (**C**) $Bi_2O_2CO_3$/CdS QDs-10 (**D**), $Bi_2O_2CO_3$/CdS QDs-15 (**E**) and $Bi_2O_2CO_3$/CdS QDs-20 (**F**).

Figure 2A shows the XRD patterns of the prepared $Bi_2O_2CO_3$, $Bi_2O_2CO_3$/CdS QDs-10 photocatalysts. The XRD characteristic diffraction peaks of $Bi_2O_2CO_3$ nanoflower well correspond

to tetragonal $Bi_2O_2CO_3$ (JCPDS card No. 41-1488) [21,22]. The XRD diffraction peaks of CdS QDs cannot be clearly distinguished through Figure 1A, indicating the low crystallinity of the prepared CdS. In the high resolution XRD pattern of CdS as shown in Figure 2B, the (111), (200) and (311) crystal planes can well correspond to the cubic phase of CdS (JCPDS 80–0019) [23,24]. For the XRD pattern of $Bi_2O_2CO_3$/CdS QDs-10, except of the characteristic diffraction peaks of $Bi_2O_2CO_3$, the diffraction peaks of CdS cannot be found due to its low crystallinity and high dispersion on the surface of $Bi_2O_2CO_3$ nanoflower.

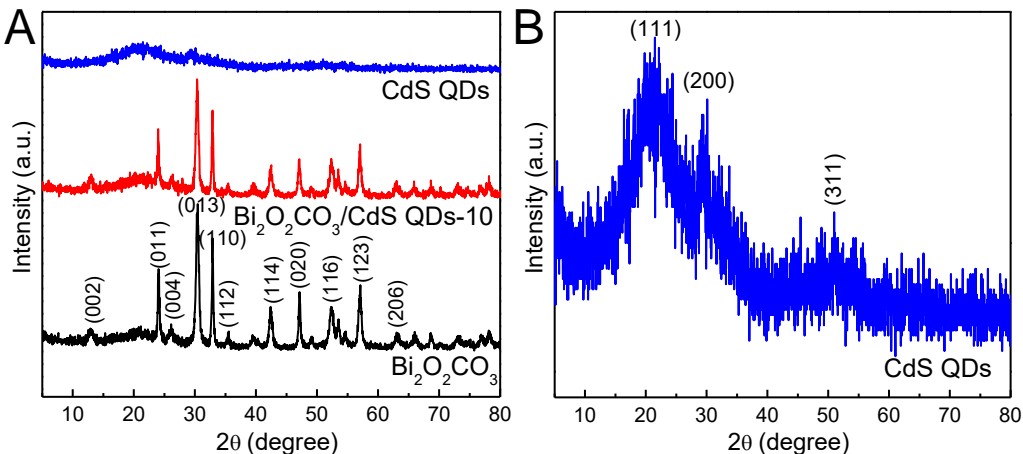

**Figure 2.** XRD patterns of the prepared $Bi_2O_2CO_3$, $Bi_2O_2CO_3$/CdS QDs-10 and CdS QDs photocatalysts (**A**); XRD curve of the pristine CdS QDs photocatalyst (**B**).

TEM characterization is further used to observe the distribution state of CdS QDs on the surface of $Bi_2O_2CO_3$, and the results are shown in Figure 3. As can be found in Figure 3A, some QDs with a size of approximately 5 nm appear on the $Bi_2O_2CO_3$ nanoflakes. With higher resolution as shown in Figure 3B, the observed lattice spacing of 0.27 nm and 0.33 nm correspond well to the (110) crystal plane of $Bi_2O_2CO_3$ and (110) crystal plane of CdS, respectively. The lattice spacing of CdS QDs cannot be observed clearly, which further proves the low crystallinity of CdS.

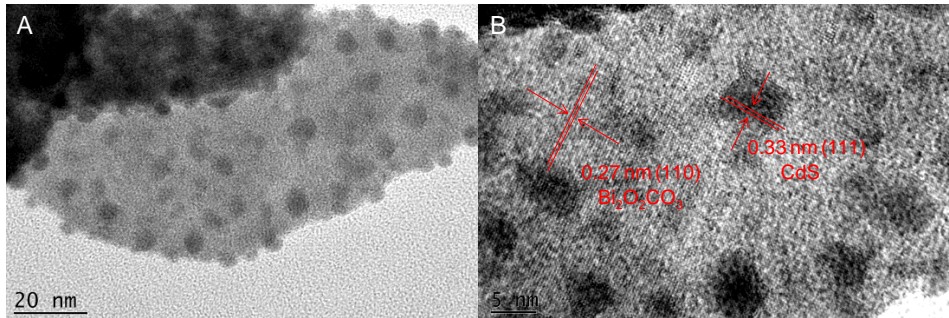

**Figure 3.** TEM images of the prepared $Bi_2O_2CO_3$/CdS QDs-10 photocatalyst: low magnification (**A**) and high magnification (**B**).

Figure 4 shows the SEM image and corresponding EDS measurement of the prepared $Bi_2O_2CO_3$/CdS QDs-10 photocatalyst. Figure 4A,B show the SEM image and corresponding EDS spectrum. The elements of C, O, Bi, S and Cd can be clearly observed, which are well consistent with the elements in $Bi_2O_2CO_3$/CdS QDs-10. The EDS elemental mapping results are displayed in Figure 4C–G. The distribution of O, Bi, S and Cd elements can be observed well corresponding to the morphology of $Bi_2O_2CO_3$/CdS QDs-10 in Figure 1A, which proves the uniform distribution of CdS QDs on the surface of $Bi_2O_2CO_3$ hierarchical nanoflower. The C element is distributed in the

whole scanning area because of the carbon conductive adhesive as the substrate in the process of SEM measurement. No other impurity elements are found in the EDS test, indicating the purity of $Bi_2O_2CO_3$/CdS QDs-10.

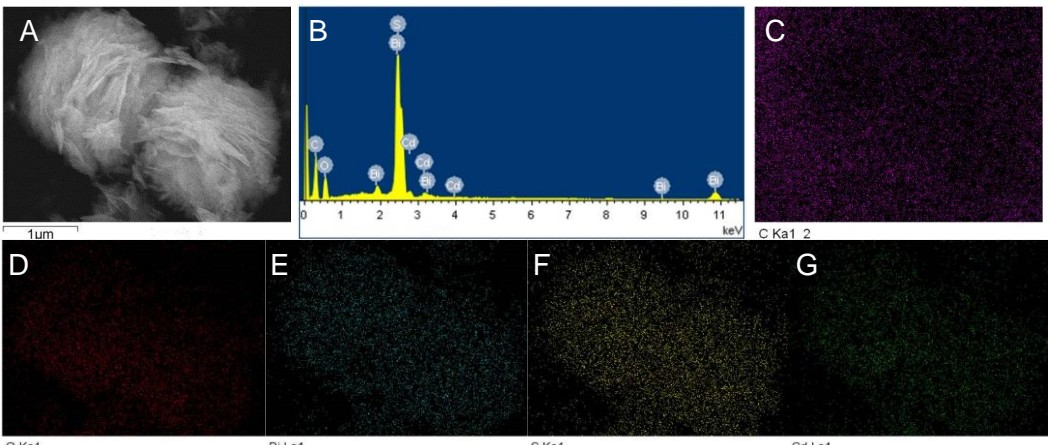

**Figure 4.** SEM image (**A**), the corresponding EDS spectrum (**B**) and EDS elemental mappings (**C**–**G**) of $Bi_2O_2CO_3$/CdS QDs-10.

The XPS measurement was further used to analyze the elemental states of the prepared $Bi_2O_2CO_3$/CdS QDs-10 photocatalyst. Figure 5 displays the XPS results of $Bi_2O_2CO_3$/CdS QDs-10. Cd, Bi, S, C and O elements can be found in the total survey spectrum as shown in Figure 5A, which well corresponds to the element composition in $Bi_2O_2CO_3$/CdS QDs-10. The bind energy peaks of C 1s at 284.82 and 288.69 eV in Figure 5B are attributed to C-C bond and carbonate ion in $Bi_2O_2CO_3$, respectively [18,25]. Figure 5C shows the O 1s peak for $Bi_2O_2CO_3$/CdS QDs-10. The XPS peak at 529.81 is ascribed to the Bi-O bond of $Bi_2O_2CO_3$. The other two peaks at 530.75 and 531.42 eV can be labeled as C-O bond and the surface hydroxyl groups adsorbed on $Bi_2O_2CO_3$/CdS QDs-10 [26]. The XPS peaks of Bi 4f at 159.17 and 164.44 eV are attributed to the $[Bi_2O_2]^{2+}$ layers in $Bi_2O_2CO_3$. Figure 5E,F present the high-resolution spectra of S 2s and Cd 3d. The XPS characteristic peak of S 2s at 225.72 eV well corresponds to $S^{2-}$, and that of Cd 3d at 405.15 and 411.85 eV are well ascribed to $Cd^{2+}$ in CdS [27,28].

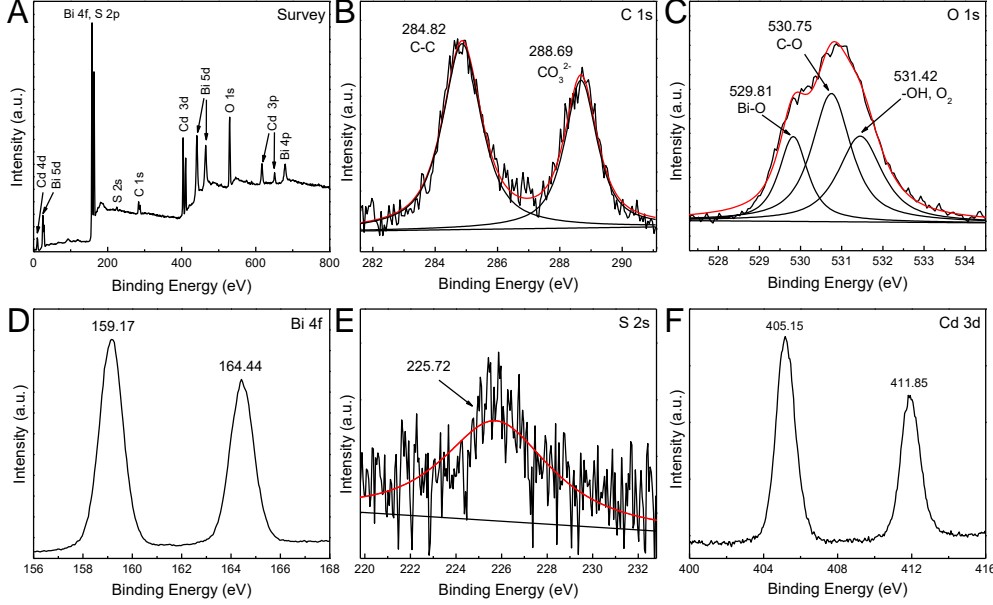

**Figure 5.** XPS spectra of the prepared $Bi_2O_2CO_3$/CdS QDs-10 photocatalyst: the total survey (**A**), C 1s (**B**), O 1s (**C**), Bi 4f (**D**), S 2p (**E**), Cd 3d (**F**) XPS core level spectra.

Based on the above results, it can be concluded that the CdS QDs decorated 3D $Bi_2O_2CO_3$ hierarchical nanoflower photocatalysts were successfully synthesized in the present work. The $Bi_2O_2CO_3$ hierarchical nanoflower can act as the carrier for the deposited CdS, and CdS QDs with low crystallinity are evenly dispersed on the surface of hierarchical nanoflower to form the heterojunction. Heterojunction photocatalysts with hierarchical structure can provide a large number of active sites for photocatalytic reaction, thus accelerating the process of photocatalytic reaction.

Figure 6A displays the photocatalytic RhB degradation performance of the prepared $Bi_2O_2CO_3$, CdS QDs and $Bi_2O_2CO_3$/CdS QDs photocatalysts under visible light illumination. Before the degradation test, the photocatalysts and RhB solution were mixed and stirred in dark for 30 min to achieve adsorption equilibrium. As can be found in Figure 6A, the adsorption capacities of the prepared photocatalysts for RhB are very weak. Under 150 min visible light illumination, $Bi_2O_2CO_3$ and CdS QDs can only degrade about 28% and 52% of rhodamine, respectively. The prepared $Bi_2O_2CO_3$/CdS QDs photocatalysts show significantly enhanced photocatalytic degradation performance. With the increase in CdS QDs deposition, the photocatalytic degradation performance of $Bi_2O_2CO_3$/CdS QDs first increased and then decreased. $Bi_2O_2CO_3$/CdS QDs-10 exhibits the best photocatalytic performance and can achieve 98% RhB degradation in 150 min under visible light illumination. $Bi_2O_2CO_3$/CdS QDs-5 has fewer heterojunctions, which are not enough to effectively catalyze the degradation of RhB. The deposition of large amount of CdS QDs will hinder the formed heterojunction from effectively degrading RhB in the photocatalytic reaction. Therefore, $Bi_2O_2CO_3$/CdS QDs-10 shows the optimum photocatalytic degradation performance. Figure 6B shows the cyclic photocatalytic RhB degradation test of $Bi_2O_2CO_3$/CdS QDs-10. After four cycles of photocatalytic RhB degradation tests, $Bi_2O_2CO_3$/CdS QDs-10 can degrade 97% of RhB after 150-min visible light illumination, indicating its good stability.

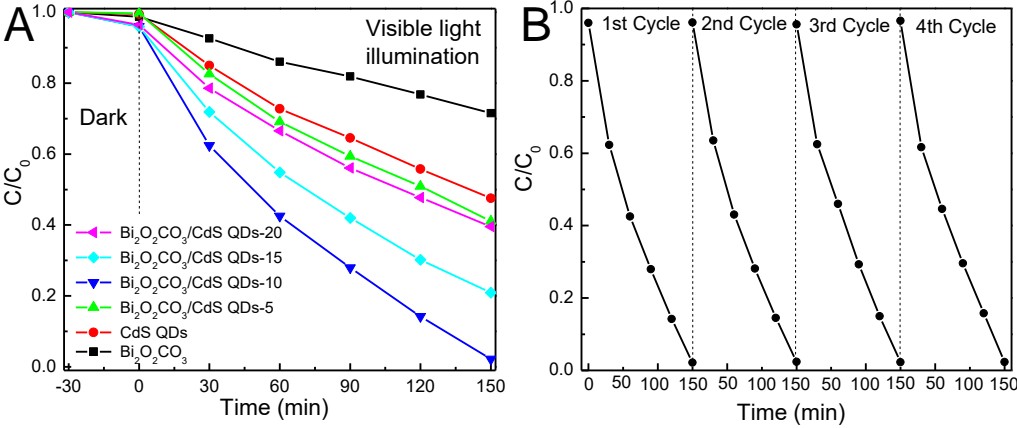

**Figure 6.** Photocatalytic RhB degradation performance of the prepared photocatalysts (**A**); the cyclic photocatalytic RhB degradation test of $Bi_2O_2CO_3$/CdS QDs-10 (**B**).

Figure 7A shows the UV-Vis absorption spectra of $Bi_2O_2CO_3$, $Bi_2O_2CO_3$/CdS QDs-10 and CdS QDs. $Bi_2O_2CO_3$ shows weaker light absorption performance compared with CdS QDs does in the range of 200–800 nm. $Bi_2O_2CO_3$/CdS QDs-10 and CdS QDs have similar light absorption thresholds, indicating the deposition of CdS QDs on the surface of $Bi_2O_2CO_3$. Compared with pure $Bi_2O_2CO_3$, $Bi_2O_2CO_3$/CdS QDs-10 displays stronger absorption and wider optical response ranges due to the deposition of CdS QDs, suggesting the enhanced light absorption performance of $Bi_2O_2CO_3$/CdS QDs than that of $Bi_2O_2CO_3$. The plot of transformed Kubelka–Munk function versus the light energy of $Bi_2O_2CO_3$ and CdS QDs are shown in Figure 7B. The Mott–Schottky measurement can be used for analyzing the band potential and carrier concentration of the prepared photocatalysts. As can be found from Figure 7C, the Mott–Schottky curves with positive slopes indicate that the prepared $Bi_2O_2CO_3$, CdS QDs and $Bi_2O_2CO_3$/CdS QDs-10 photocatalysts are n-type semiconductors. The flat band potentials of $Bi_2O_2CO_3$ and CdS QDs are obtained as −0.36 and −0.62 eV, and that of $Bi_2O_2CO_3$/CdS QDs-10 is

−0.53 eV, indicating that the formation of heterojunction makes the band of $Bi_2O_2CO_3$ and CdS QDs bend. Figure 7D shows the electrochemical impedance spectroscopy of $Bi_2O_2CO_3$, CdS QDs and $Bi_2O_2CO_3$/CdS QDs-10. A significantly reduced arc radius of $Bi_2O_2CO_3$/CdS QDs-10 indicating that $Bi_2O_2CO_3$/CdS QDs-10 has a significantly improved photogenerated carriers transfer efficiency in the process of photocatalytic reaction. Figure 7E shows the photoluminescence spectra of $Bi_2O_2CO_3$, CdS QDs and $Bi_2O_2CO_3$/CdS QDs-10. The intensity of the fluorescence peak of $Bi_2O_2CO_3$/CdS QDs-10 is significantly reduced compared with those of $Bi_2O_2CO_3$ and CdS QDs, indicating that the $Bi_2O_2CO_3$/CdS QDs effectively inhibits the recombination of photogenerated electrons and holes, thus enhancing the photocatalytic performance. Nitrogen adsorption-desorption isotherm measurements are performed to analyze the specific surface area and pore structure distribution of $Bi_2O_2CO_3$ and $Bi_2O_2CO_3$/CdS QDs-10. As shown in Figure 7F, $Bi_2O_2CO_3$ and $Bi_2O_2CO_3$/CdS QDs-10 conform to the type-IV isotherm law, indicating that the prepared photocatalysts contain the slit-shaped mesopore structure. As shown in the inserted table in Figure 7F, the calculated BET surface area of $Bi_2O_2CO_3$/CdS QDs-10 is slight lower than that of $Bi_2O_2CO_3$ due to the deposition of CdS QDs. The $V_{pore}$ and $d_{pore}$ of $CdS/Co_9S_8$-0.2 can be obtained as 0.13 $cm^3$/g and 23.79 nm, respectively.

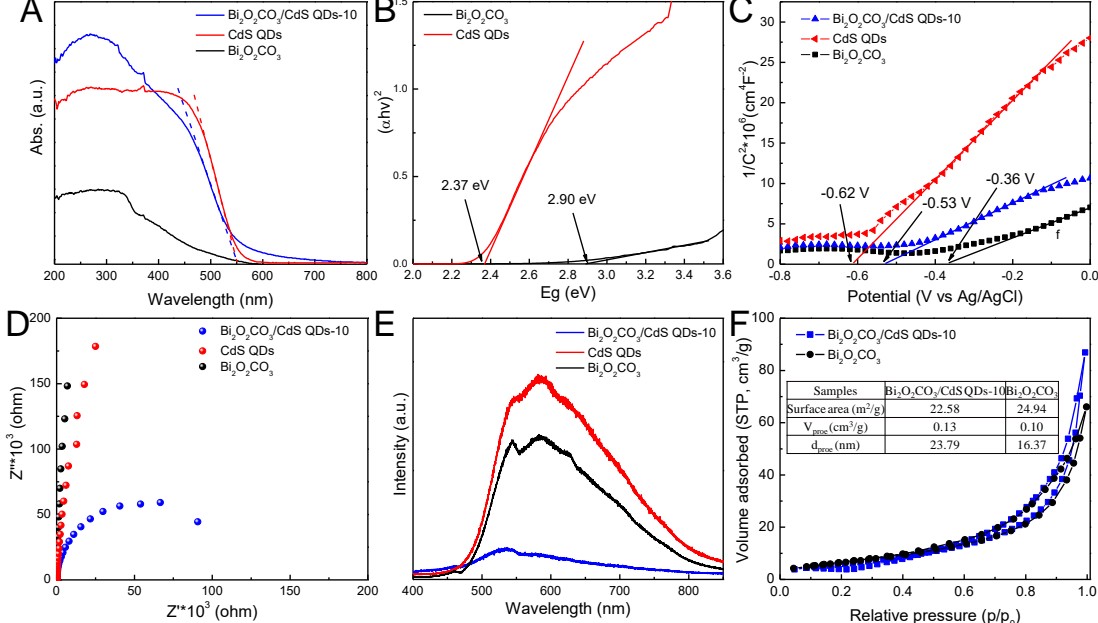

**Figure 7.** UV–Vis absorption spectra of $Bi_2O_2CO_3$, CdS QDs and $Bi_2O_2CO_3$/CdS QDs-10 (**A**); the plot of transformed Kubelka–Munk function versus the light energy of $Bi_2O_2CO_3$ and CdS QDs (**B**); Mott–Schottky plots (**C**), electrochemical impedance spectroscopy (**D**) and Photoluminescence spectra (**E**) of $Bi_2O_2CO_3$, CdS QDs and $Bi_2O_2CO_3$/CdS QDs-10; Nitrogen adsorption-desorption isotherms (The inset is the inserted table of the calculated BET surface areas, $V_{pore}$ and $d_{pore}$) of $Bi_2O_2CO_3$ and $Bi_2O_2CO_3$/CdS QDs-10 (**F**).

The photocatalytic mechanism of $Bi_2O_2CO_3$/CdS QDs-10 is further analyzed by adding different scavengers for the photogenerated holes (EDTA-$Na_2$, 5 mM), the ·OH (isopropyl alcohol, IPA, 5 mM) and the $·O_2^-$ (P-benzoquinone, BQ, 5 mM) into the RhB solution in the photocatalytic process. As displayed in Figure 8, the photocatalytic performance shows slightly reduced with adding 5 mM IPA to eliminate the ·OH. However, when 5 mM EDTA-$Na_2$ or BQ are added into the RhB solution to eliminate the photogenerated holes or $·O_2^-$, the photocatalytic degradation performance is significantly decreased under visible light illumination. The results indicate that the photogenerated holes and $·O_2^-$ are the main active groups for the photocatalytic degradation for RhB.

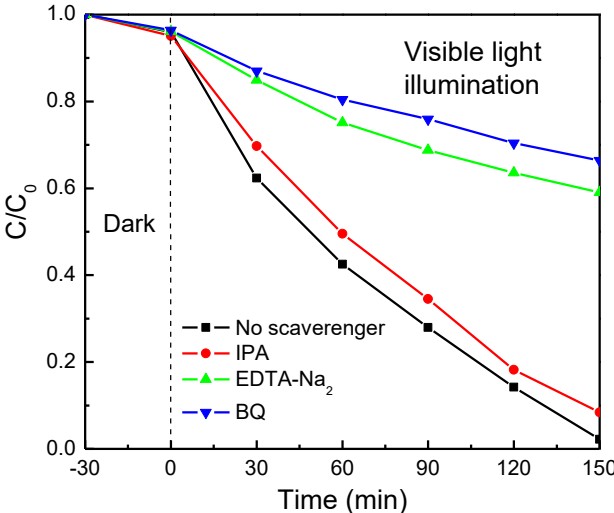

**Figure 8.** Photocatalytic RhB degradation curves of $Bi_2O_2CO_3$/CdS QDs-10 containing with different scavengers.

Figure 9 shows the schematic illustration of the enhanced photocatalytic degradation performance of $Bi_2O_2CO_3$/CdS QDs-10 under visible light illumination. CdS QDs are uniformly dispersed on the surface of $Bi_2O_2CO_3$ hierarchical nanoflower to form the heterojunction. Under visible light illumination, the photogenerated electrons and holes are excited to the conduction band (CB) and valence band (VB) of $Bi_2O_2CO_3$ and CdS QDs, respectively. Driven by the heterojunction electric field, the photogenerated electrons formed on the CB of CdS can quickly transfer to the CB of $Bi_2O_2CO_3$ and catalyze the absorbed $O_2$ to form $\cdot O_2^-$. Concurrently, the photogenerated holes of $Bi_2O_2CO_3$ flow to the VB of CdS and transfer to the surface of photocatalyst rapidly. In this way, the photocarriers can be separated quickly and efficiently in the photocatalytic reaction due to the $Bi_2O_2CO_3$/CdS heterojunction. Furthermore, the 3D $Bi_2O_2CO_3$ hierarchical nanoflowers provide a large number of reactive sites, which can cooperate with the formed $\cdot O_2^-$ and photogenerated holes to accelerate the photocatalytic reaction process, thus enhancing the photocatalytic degradation performance of $Bi_2O_2CO_3$/CdS QDs.

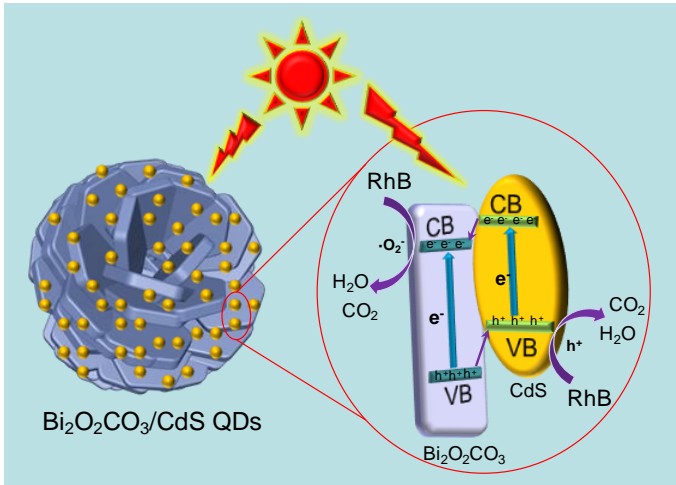

**Figure 9.** Schematic illustration of the enhanced photocatalytic degradation performance of $Bi_2O_2CO_3$/CdS QDs-10.

## 3. Experimental Section

### 3.1. Preparation of the Series of Photocatalysts

The $Bi_2O_2CO_3$ hierarchical nanoflower was prepared by a hydrothermal method. Amounts of 0.97 g $Bi(NO_3)_3 \cdot 5H_2O$, 1.76 g sodium citrate, 0.48 g urea and 1 g polyvinylpyrrolidone (PVP) were dispersed in 60 mL deionized water under stirring for 45 min. The mixed solution was transferred into a 100 mL Teflon-lined autoclave, and heated in an oven at 180 °C for 12 h. The precipitate was washed 5 times and dried at 60 °C to obtain the $Bi_2O_2CO_3$ hierarchical nanoflower.

The preparation of CdS QDs follows the following steps: First, 4.00 g $CdCl_2 \cdot 2.5H_2O$ was dispersed in 200 mL deionized water with the thioglycollic acid (TGA) as stabilizer. Then the pH of the solution was adjusted to 10 by using 1 M NaOH. Then 10 mL solution containing 0.84 g $Na_2S \cdot H_2O$ was dropped into the $CdCl_2$ solutions, and the mixed solution was stirred at 60 °C for 60 min. The obtained solution was centrifuged, and CdS QDs were extracted and washed with deionized water for 5 times. CdS QDs solution was obtained by redispersing the obtained CdS QDs into deionized water.

The $Bi_2O_2CO_3$/CdS QDs was prepared as the following method: First, 0.3 g $Bi_2O_2CO_3$ was dispersed into 20 mL deionized water containing 0.015 g, 0.03 g, 0.045 g and 0.06 g CdS QDs, respectively. After stirring for 60 min, 10 mL acetone was added into the above mixed solution and stirring for 24 h. The obtained precipitates were washed 5 times with deionized water and dried at 80 °C overnight. The precipitates were then calcined at 300 °C for 1 h and labeled as $Bi_2O_2CO_3$/CdS QDs-5, $Bi_2O_2CO_3$/CdS QDs-10, $Bi_2O_2CO_3$/CdS QDs-15, $Bi_2O_2CO_3$/CdS QDs-20, respectively. The numbers 5, 10, 15 and 20 represent the mass percentage of CdS in $Bi_2O_2CO_3$, respectively.

### 3.2. Characterization of the Prepared Photocatalysts

The morphology of the prepared $Bi_2O_2CO_3$, CdS QDs and $Bi_2O_2CO_3$/CdS QDs photocatalysts was observed by field emission scanning electron microscopy (FE-SEM, Ultra 55, Zeiss, Germany) and field emission transmission electron microscope (FE-HRTEM, Tecnai G2 F20, FEI Company, Hillsboro, OR, USA). The crystal structures of the prepared photocatalysts were identified by using an X-ray diffractometer (XRD, D/max-500, Rigaku Co., Tokyo, Japan). The element composition, distribution and Chemical states of $Bi_2O_2CO_3$/CdS QDs-10 are characterized by an X-ray energy dispersive spectrometer (EDS, Oxford, UK) and an X-ray photoelectron spectroscopy (XPS, ESCALAB 250Xi, England American Somerfly Technology Company, Schenectady, NY, USA). The optical absorption properties of the prepared $Bi_2O_2CO_3$, CdS QDs and $Bi_2O_2CO_3$/CdS QDs-10 photocatalysts was analyzed by a UV-visible diffuse reflectance spectrophotometer (U-41000; HITACHI, Tokyo, Japan). $BaSO_4$ was used as the reference material in the process of optical absorption performance tests.

### 3.3. Photocatalytic Degradation Performance and Electrochemical Measurement

An amount of 100 mg of the prepared photocatalysts was dispersed into 100 mL of RhB aqueous solution (10 mg·L$^{-1}$) in the photocatalytic degradation tests. The mixture was kept for stirring for 30 min in the dark before the tests. A 300 W Xe lamp (PLS-SXE300, Beijing Changtuo Co., Ltd., Beijing, China) was applied as the light source, and a 420 nm cut-off filter to remove ultraviolet light was used to obtain the visible light. The visible light illumination intensity is controlled at 250 mW·cm$^{-2}$ in the photocatalytic degradation tests. Circulating water system is used to maintain the temperature of the degraded solution at 25 °C. In the process of photocatalytic degradation stability test, the photocatalyst was recovered by centrifugation and fully dried before the next cycle photocatalytic degradation of RhB solution. The degradation conditions were the same as before, and the number of cycles was 4.

The electrochemical test was carried out by the CHI660D electrochemical workstation (Shanghai Chenhua Instrument Co., Ltd., Shanghai, China). The typical three electrode system (the preparation electrode is the working electrode, Pt is the counter electrode and Ag/AgCl is the reference electrode) is used in the test process, which is carried out in 0.1 M $Na_2SO_4$ solution. The test technology used in the Electrochemical Impedance Spectroscopy test was AC impedance. The applied

bias voltage was 0 V and the frequency range was $10^5$–$10^{-2}$ Hz. The Mott–Schottky test used the impedance potential technology, with a voltage range of $-0.8$–$0$ V and a frequency of 1000 Hz. One point was taken every 20 mV during the scanning process.

## 4. Conclusions

In summary, a novel 3D $Bi_2O_2CO_3$/CdS QDs hierarchical nanoflower heterojunction photocatalyst has been successfully prepared to explore the enhanced photocatalytic degradation performance for RhB. CdS QDs are uniformly dispersed on the surface of $Bi_2O_2CO_3$ hierarchical nanoflower to form the heterojunction. The prepared $Bi_2O_2CO_3$/CdS QDs photocatalysts show significantly enhanced photocatalytic degradation performance compared with $Bi_2O_2CO_3$ and CdS QDs, and $Bi_2O_2CO_3$/CdS QDs-10 exhibits the best photocatalytic performance which can achieve 98% RhB degradation in 150 min under visible light illumination. The establishment of the heterojunction can improve the efficiency of photocarrier transfer and restrain the recombination of photogenerated electrons and holes. In addition, the photogenerated holes and formed $\cdot O_2^-$ are the main active groups for the degradation of RhB in the photocatalytic process. The 3D $Bi_2O_2CO_3$ hierarchical nanoflower can provide a large number of reactive sites, which can cooperate with the formed $\cdot O_2^-$ and photogenerated holes to accelerate the photocatalytic reaction process, thus enhancing the photocatalytic degradation performance of $Bi_2O_2CO_3$/CdS QDs.

**Author Contributions:** Z.J.: literature search, data collection, figures, data analysis, data interpretation, writing. H.W.: literature search, conception or design of the work; data analysis, Drafting the work or revising it critically. X.S.: literature search, analysis and interpretation of data for the work, Drafting the work or revising it critically. All authors have read and agreed to the published version of the manuscript.

**Funding:** This research was funded by the Natural Science Foundation of Shandong Province (ZR2017MEM018).

**Conflicts of Interest:** The authors declare no conflict of interest.

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
