# Peer review of "A Novel CdS Quantum Dots Decorated 3D Bi2O2CO3 Hierarchical Nanoflower with Enhanced Photocatalytic Performance"

_catalysts, doi:10.3390/catal10091046_

Round 1

Reviewer 1 Report

The authors present experimental studies on Bi2O2CO3/CdS QDs hierarchical nanoflower heterojunctions to investigate their ability to act as photocatalyst. The authors used several techniques to demonstrate the structural properties as well as the performance. The results are conclusive that one of their systems, Bi2O2CO3/CdS QDs-10 (0.03g CdS QDs), performs well as photocatalysts in comparison to pure Bi2O2CO3 or CdS QDs as well as for Bi2O2CO3/CdS QDs with other concentration of CdS.

The study is interesting and shows new results, however, some points need to be clarified and/or better explained:

  1. In the study four different heterojunctions concentrations of CdS QDs are fabricated and only one of them (Bi2O2CO3/CdS QDs-10) is discussed in all details. Interesting is also the information about the other concentrations in respect to e.g. the results from EDS, UV-Vis absorption and electrochemical impedance spectroscopy. Even though the performance is not suitable to act as a photocatalyst, it is still interesting how the concentration change the properties.
  2. The information of the XRD study have no real significance since it shows mainly that one cannot detect any signature of CdS QDs in Bi2O2CO3/CdS QDs-10. Moreover, unclear is why the statistics is not improved by e.g. binning of the points (density of the points in 2theta is not required) and longer measurement times that could possibly help to see some hints of the QDs. The reflections indicated in Figure 1B (220) and (311) are very undefined on the level of noise/background. The (111) seems as if it can be also seen in the XRD of Bi2O2CO3 and thus might have other origin. Please clarify these points or provide measurements with better statistics if possible.
  3. The crystallographic planes indicated in the TEM images (Figure 3) are hard to see for the CdS QDs (as well as the red color of the writing) in the resolution provided in the picture and the reader can only guess. A picture with better resolution (magnified?) should be provided that show the lattice planes for the CdS lattice planes clearer or the statement has to be modified.
  4. The photocatalytic RhB degradation measurements demonstrates that Bi2O2CO3/CdS QDs-10 has the best performance and is explained by the authors with the “right” amount of heterojunctions in comparison with the samples of more or less CdS QDs. The explanations are only qualitative and of more speculative nature even though plausible. Here, any other indications or, if feasible, theoretical estimation/calculations supporting the statement would be very helpful to give it more insight, e.g. how the performance is linked with the concentration. Besides the preparation of the sample where the amount of CdS is varied added into the solution, the question is whether the concentration levels in the prepared samples is similar. Is it possible to determine/estimate the “real” occupation of CdS (by quantitaive EDS?)?
  5. Missing in the paper is also a statement how well the Bi2O2CO3/CdS QDs performs in respect to other photocatalyst systems with similar heterojunctions. It would be important to judge about the potential of the system as a future photocatalyst.
  6. Besides these points concerning the content, there are also some other aspects that need to be improved:
    1. The text is not well written and contains many grammar errors and misspellings. Short cuts are used without naming the full name should be avoided (e.g. line 44 SPR effect), even though they might be common to the community. This needs to be improved.
    2. For the common reader it might be not totally clear what the Motto-Schottky test is and what exactly is measured with it.
    3. Line 122-123: it is referred to figure 1A (“seen clearly seen through figure 1A”) that makes no sense how it is written and needs reformulated for better understanding.

In the present form, I do not recommend the manuscript for publication without clarifying these aspects.

Reviewer 2 Report

Ji et al. here present an interesting study on the synthesis, characterization and testing of QD-nanoflower composite nanostructures for the photocatalytic degradation of rhodamine B. The abstract and introduction provide a clear background, and convey well the rationale of the work undertaken. The experimental section appears to give enough information for the experiments to be repeated independently. The results are clearly presented in a logical order. I detail below my thoughts on this manuscript.

The name Hongwei Wang I presume should not be separated by a comma.

It would be better to mention the targeted substrate for the catalyzed reaction in the abstract.

RhB should be defined at first mention.

In the Experimental Section… “0.3 g Bi2O2CO3 was dispersed
75 into 20 mL solution containing 0.015 g,0.03 g,0.045 g and 0.06 g CdS QDs, respectively”… was this a buffered aqueous solution? Or pure water?

Line 77, “obtained precipitates were washed 5 times”… washed with what?

Line 78, “Bi2O2CO3/CdS QDs-5, Bi2O2CO3/CdS QDs-10, Bi2O2CO3/CdS QDs-15, Bi2O2CO3/CdS QDs-20”… it should be made clear what the numbers 5, 10, 15, 20 refer to in this naming convention.

Line 91, “100 mL of RhB solution”… pure water?

Line111, The pristine CdS agglomerates into the nanoparticle with the size of dozens of nanometers shown in Figure 1B”… does this mean that the CdS agglomerates are dozens of nanomaters? The phrasing of this sentence is unclear.

Line 112, “dozens of nanometers”… this is too imprecise.

Figure 1, why are the CdS agglomerates clearly visible in panel B, but not in the other images C-F? This is confusing as the amount of CdS should be increasing through these different samples, right? This is especially confusing as later in the text, the authors claim that the samples in panel E and F have too many CdS QDs deposited on their surface to exhibit efficient photocatalysis.

Line 123, “low crystallinity of the prepared CdS”… I think this is to be expected from QDs synthesized by an aqueous route. What affect does this poor CdS quality have on the following reaction studies?

Figure 5A, there are no error bars in this plot. Were the experiments performed only once?

Line 183, “Bi2O2CO3/CdS QDs-10 displays the strong absorption over the entire spectrum, suggesting the best photocatalytic performance”… I don not agree with this statement. The absorbance reaches near zero at 600 nm, and is decaying from 300 nm upwards.

Figure 6A, the absorption behavior below ca. 370 nm looks very questionable. What material of absorption cuvette was used? Also, why is the absorption behavior below 420 nm relevant, as the authors have used a 420 nm cut-off filter in the illumination studies? Further, as the pure CdS QD and nano flower samples do not show any long wavelength (>600 nm) absorption, is it right to expect that the mixed sample would? It seems that by 800 nm the absorption would be near zero. Is there a problem with the baseline of this spectrum?

The appears to be no excitonic peak in the CdS absorption spectrum. Did the authors measure a photoluminescence spectrum of these QDs? Do they exhibit any fluorescence? I’m concerned by the low quality of the CdS QDs and the effect that this would have on the proposed reaction mechanism. I think this should be addressed in the text.

To summarize, my main concerns with the manuscript are the confusion over the visibility of CdS aglgomerates in Figure 1, and over the quality of the CdS QDs and the affect of this on the proposed reaction mechanism. Both of these points, along with the minor comments above, need to be addressed before publication can be considered.

Reviewer 3 Report

The manuscript (catalysts-862587) by Zhilin Ji, et al. presents investigation of CdS quantum dots decorated 3D Bi2O2CO3 hierarchical nanoflower photocatalyst. The authors investigate structural, optical and electronic properties of the Bi2O2CO3/CdS QDs with different amount of CdS nanoparticles dispersed on the surface of Bi2O2CO3 nanoflowers, and compare it to the pristine Bi2O2CO3 and CdS QDs. They use a combination of XRD, SEM/EDS, TEM, and UV-Vis and electrochemical impedance spectroscopy spectroscopy to characterize structural and electronic properties responsible for the observed differences in the photocatalytic performance. They identified the optimal loading of CdS QD that gives best photocatalyst performance (Bi2O2CO3/CdS QDs-10).

The paper is clearly written and figures represent well the experimental results. The conclusions are supported by experimental results. However additional experimental evidences would be valuable to additionally support the conclusions: the comparison of the specific surface areas of the photocatalysts and XPS analysis of the electronic bands in the material. Some additional relevant references should be added.

To my opinion the paper merits the publication after the revision mentioned above.

Round 2

Reviewer 1 Report

The authors replied to my points and most of the responses are satisfactory. Moreover, they added some additional results that made the statements about the good performance of the heterojunctions Bi2O2CO3/CdS QDs-10 as a photocatalyst more solid. In principle, the paper can be recommened for publication, however, some aspects should be still improved or at least modified/corrected.

  1. The authors did not completely answered my concerns that I expressed with my second point concerning the interpretation of the diffraction data. From given answer, the confirmation of the tetragonal formation of Bi2O2CO3 seems to be an important result that is also indicated in the new version. For the diffraction signal of the CdS QDs photocatalysts, it is still very questionable to attribute the weak peaks to any crystallographic signature. A) In the here presented statistics the (220) peak cannot be really seen and also the other peaks could have been measured (presented) at least clearer with longer measurement times and/or with better binning to enhance the statistics (the chosen resolution (2theta-steps) is not required and lead to a noisy signal). B) The broad intensity distribution that is attributed to the (111) peak of CdS QDs (blue curve) seems to be also present for the purely prepared Bi2O2CO3 where no CdS QDs is present. It arises the question whether that is instrumental background. This needs to be clarified. Even though it is not a major point, the interpretation of data should be clear.
  2. The answer for Point 3 of my first review concerning the TEM images is still not completely convincing since it more or less just repeats the statements made before in the first version. For sure, the lattice spacing of 0.27 nm of Bi2O2CO3 (110) can be identified very clearly. The drawn 0.33 nm lattice of CdS (111) spacing, on the other hand, is still not convincing, even if looking very carefully on the provided new graph. It seems that the dark spots that are attributed to the CdS QDs have no crystal structure that can be identified clearly in the image which is maybe no surprise due to their low crystallinity. If the authors want to stick on their presentation and claiming such a spacing, it would be necessary to present/explain that clearer.
  3. The authors improved the English in the text, however, there are still typos and grammatical mistakes in the text that makes some parts harder to read, particularly the newly added text. Articles are sometimes missing and inconsistencies exists in stating number and units (e.g. 150 min .or 150-min). Moreover, it would improve the readability of the text to give the full name of short-cuts (e.g. XPS) and, as long as it can be done in one, two sentence (can be later in a separate section), what the method does.

Reviewer 2 Report

I have reviewed all corrections. I am not convinced by the answer to point 14, as the reply did not address the questions that I was asking, or explain the logic very well. I suspect the absorption spectra below 370 nm are not correct, and would need to be convinced that there are accurate.

Specifically...

1) the absorption behavior below ca. 370 nm looks unrealistic. Why is this?

2) what material of cuvette was used?

3) what correction was made to the baseline?

4) The reply "I used BaSO4 as the reference material to test the light absorption performance of the prepared photocatalysts" needs to be clarified further. I cannot find mention of this in the manuscript.

Further, the PL spectra in Fig 7E look strange to me, and they are not well explained.

1) How are the samples normalized to each other? Are they normalized to the same concentration at the the excitation wavelength? Or are the molar amounts of the two components consistent before and after mixing? If not, how are they comparable? This needs to be explained.

2) Why do the CdS QDs and Bi2O2CO3 spectra exhibit the the same features? Would this really be expected, given they are different materials?

3) Why does the PL of the CdS QDs look very different to the standard PL of QDs?

Round 3

Reviewer 2 Report

The further revisions are satisfactory and I would now recommend the manuscript for publication.